# Symbioflor2^®^
*Escherichia coli* Genotypes Enhance Ileal and Colonic Gene Expression Associated with Mucosal Defense in Gnotobiotic Mice

**DOI:** 10.3390/microorganisms8040512

**Published:** 2020-04-03

**Authors:** Unai Escribano-Vazquez, Claudia Beimfohr, Deborah Bellet, Muriel Thomas, Kurt Zimmermann, Philippe Langella, Claire Cherbuy

**Affiliations:** 1Micalis Institute, INRAE, AgroParisTech, Université Paris-Saclay, 78350 Jouy-en-Josas, France; unai.evo@gmail.com (U.E.-V.); deborah.bellet@hotmail.fr (D.B.); muriel.thomas@inrae.fr (M.T.); philippe.langella@inrae.fr (P.L.); 2Vermicon AG, Emmy-Noether Strasse 2, 80,992 München, Germany; beimfohr@vermicon.com; 3SymbioGruppe GmbH & Co KG, 35745 Herborn, Germany; Kurt.Zimmermann@symbio.de; 4SymbioPharm GmbH, 35745 Herborn, Germany

**Keywords:** *Escherichia coli*, Symbioflor2^®^, intestine, innate immune response, gnotobiotic mice

## Abstract

Symbioflor2^®^ is a probiotic product composed of six *Escherichia coli* genotypes, which has a beneficial effect on irritable bowel syndrome. Our objective was to understand the individual impact of each of the six genotypes on the host, together with the combined impact of the six in the compound Symbioflor2^®^. Gnotobiotic mice were mono-associated with one of the six genotypes or associated with the compound product. Ileal and colonic gene expression profiling was carried out, and data were compared between the different groups of gnotobiotic mice, along with that obtained from conventional (CV) mice and mice colonized with the probiotic *E. coli* Nissle 1917. We show that Symbioflor2^®^ genotypes induce intestinal transcriptional responses involved in defense and immune mechanisms. Using mice associated with Symbioflor2^®^, we reveal that the product elicits a balanced response from the host without any predominance of a single genotype. The Nissle strain and the six bacterial genotypes have different effects on the intestinal gene expression, suggesting that the impacts of these probiotics are not redundant. Our data show the effect of the Symbioflor2^®^ genotypes at the molecular level in the digestive tract, which further highlights their beneficial action on several aspects of intestinal physiology.

## 1. Introduction

Probiotics are commonly defined as live non-pathogenic microorganisms that confer health benefits on the host, when administered in adequate amounts [1]. Although the most frequently used probiotics are lactic bacteria, mainly the *Lactobacillus* and *Bifidobacterium* genera, some *Escherichia coli* strains were also shown to have beneficial effects on human health. These strains constitute the basis of at least three commercially available probiotic products, known under the commercial names Mutaflor^®^, Symbioflor2^®^, and Colinfant^®^ [2,3]. *E. coli* Nissle 1917 (Nissle), the active component of Mutaflor^®^, is one of the most documented probiotics for therapeutic applications. A number of previous studies revealed the benefit of this strain in humans as an oral treatment for intestinal disorders including ulcerative colitis (UC) [4] and irritable bowel syndrome (IBS) [5,6]. *E. coli* Nissle was shown to be as effective as 5-aminosalicylic acid in three large studies [7,8,9], and its efficacy in maintaining remission in UC is also recognized by the ECCO (European Crohn’s and Colitis Organisation) guidelines [10].

In contrast to Mutaflor, which is only composed of the Nissle strain, Symbioflor2^®^ contains six *E. coli* genotypes called G1/2, G3/10, G4/9, G5, G6/7, and G8. All genotypes were originally isolated from a single healthy human donor and are present at the levels of 20%, 20%, 20%, 10%, 20%, and 10%, respectively, in the final product. Clinical trials showed that Symbioflor2^®^ is effective in reducing symptoms of IBS patients in adults [11] and children [12]. Furthermore, fecal β-defensin-2 (hBD-2), which is a major component of the innate immune system, increased after three weeks of Symbioflor2^®^ intake by healthy volunteers and remained elevated nine weeks after treatment arrest [13]. *In vitro* experiments reported that one of the Symbioflor2^®^ genotypes increases hBD-2 production to a level similar to that observed with the Nissle strain [13]. Interestingly, comparison of the genome of 4 Symbioflor2^®^ genotypes revealed that they share more genes with the *E. coli* model commensal strain MG1655 than with an enteropathogenic strain [14].

The aim of this study on gnotobiotic mice was (i) to investigate the effect of each single genotype contained in Symbioflor2^®^ and of the combined probiotic product on small intestinal and colonic gene expression profile, and (ii) to compare gene expression profiles obtained with each Symbioflor2^®^ genotype with those obtained in the Nissle mono-associated and conventional (CV) mice. Gene expression profiling was carried out by high-throughput QPCR with a selection of candidate genes mainly associated with intestinal mucosal defense. The six bacterial genotypes of Symbioflor2^®^ increase the expression of genes involved in several cell functions, especially in mucosal defense, such as enzymes involved in the turnover of reactive oxygen species (ROS), *duox2* and *duoxa2,* and antimicrobial peptide production, *reg3β*, *reg3γ*, *ang4*, and *pla2g2a*. Our data reveal that the Nissle strain and the six bacterial strains of Symbioflor2 have a different impact on the intestinal gene expression profile, mainly in the colon. This suggests that these probiotics are not redundant and can each exert an independent beneficial effect on gut health. We also show that the product targets a core set of genes. Thus, the probiotic product elicits a balanced response from the host without any predominance of a single genotype, revealing that the balance between strains established in the preparation is preserved upon ingestion.

## 2. Materials and Methods

### 2.1. Ethics Concerning Animal Experimentation

Experiments involving gnotobiotic mice were performed at the Anaxem platform of the MICALIS Institute (INRA, Jouy-en-Josas, France). The Anaxem facility is accredited by the French “Direction Départementale de la Protection des Populations (DDPP78)” accreditation number A78-322-6. Experiments involving CV mice were performed at the IERP facility (INRA, Jouy-en-Josas, France), which is accredited by the French “Direction Départementale de la Protection des Populations (DDPP78)”, accreditation number DDPP-VET-13-0124. All procedures involving animal experimentation were carried out according to the European guidelines for the care and use of laboratory animals under the authority of a license issued by the French Veterinary Services (authorization number 78–122 specific to CC) and were approved by the French “Ministère de l’Enseignement Supérieur et de la recherche” (authorization number APAFIS#3441-2016010614307552). The Symbioflor2^®^ product and the six *E. coli* genotypes contained in Symbioflor2^®^ (G1/2, G3/10, G4/9, G5, G6/7, G8) were provided by Symbiopharm. These *E. coli* isolates originated from the stool of one healthy individual in Germany in 1954 and together comprise the product Symbioflor2^®^ DSM 17252.

### 2.2. Establishment of Gnotobiotic Mice

All germ-free (GF) mice (seven- to eight-week-old males, C57Bl/6) were purchased from the GF rodent breeding facilities of the CNRS-TAAM (transgenesis, archiving and animal models) center (Orléans, France). They were delivered to Anaxem under sterile conditions and immediately transferred into the experimental isolator. After reception, the GF mice were left undisturbed for eight days before starting the experiment, and their GF status was verified by microscopic observation of fresh feces and by culturing fecal material on various bacterial culture media. The mice received a standard diet, R03-40 (Scientific Animal food and Engineering, Augy, France), sterilized by gamma irradiation at 45 kGy, for the duration of the experiment.

#### 2.2.1. Mouse Inoculation

GF mice were inoculated once by oral gavage with a volume of 100 µL of bacterial suspension. Inocula were prepared as follows: the day before inoculation, each of the six *E. coli* genotypes and the Nissle strain were cultured overnight on LB (Luria Bertani) medium. The fresh overnight cultures were centrifuged for 15 min at 3000× *g* and the bacterial pellet was resuspended in phosphate-buffered saline (PBS) at a concentration of 10^8^ bacteria/100 µL. Freshly prepared bacteria were immediately introduced into the isolator for mouse inoculation. For inoculation with Symbioflor2^®^, we gave 5 × 10^7^ viable bacteria/mouse taken from the probiotic product.

#### 2.2.2. Animal Experimentations

There were 10 groups of mice with 10 mice to each group. We used seven groups of monoxenic mice (inoculated with G1/2, G3/10, G4/9, G5, G6/7, G8, and Nissle), one group of mice inoculated with Symbioflor2^®^ (Symb group), consisting of the mixture of all six genotypes, one group of GF mice, and one group of CV. All gnotobiotic mice were euthanized by cervical dislocation 21 days after inoculation, i.e., at the age of 11–12 weeks. CV mice were matched with gnotobiotic groups in term of strains (C57Bl/6), diet, and age at sacrifice (11–12 weeks). Cecal contents were recovered, weighed, and immediately stored at −80 °C to quantify *E. coli* colonization levels. Ileum and colon were removed, flushed with PBS, immediately frozen in liquid nitrogen, and pulverized using a Bessman tissue pulverizer under liquid nitrogen conditions. Samples were stored at −80 °C until RNA extraction.

### 2.3. Monitoring of E. coli Colonization Levels in Cecal Samples of Monoxenic Mice by QPCR

Total DNA was extracted from cecal sample contents, as described previously [15]. *E. coli* were quantified by Real-time QPCR analysis using primers targeting *E. coli* 16S ribosomal RNA (rRNA) genes as described in Reference [15]. Standard curves generated from 10-fold serial dilutions of *E. coli* strain DNA samples were used for quantification.

### 2.4. Quantification of Colonization Level of E.coli Genotypes in Mice Inoculated with Symbioflor2^®^

In mice inoculated with Symbioflor2^®^, the colonization level of each of the *E. coli* genotypes was quantified by cultivation and subsequent multiplex PCR using grown colonies from the cecal content of gnotobiotic mice. Strain-specific probes, which enabled the specific detection of *E. coli* Symbioflor2^®^ components, were designed from the genome sequences. However, no specific PCR primer sets could be found to detect the genotypes G1-2, G6-7, and G8 individually. Thus, a single primer pair was developed to detect these three genotypes together. Three other specific primer pairs were designed to detect the genotypes G3-10, G4-9, and G5, individually. The sequences of the forward and reverse primers are given in the Appendix A.

To quantify the genotypes’ abundance, cecal samples diluted in 0.9% NaCl were plated on nutrient medium (peptone 5,0 g/L, meat extract 3,0 g/L, agar 15 g/L; pH 7.0) and cultivated aerobically for 18–20 h at 37 °C. PCRs to confirm the identity of colonies were carried out as follows: DNA for each reaction was extracted from a colony by using a sterile toothpick. Then, the toothpick was vortexed in 5 µL of sterile water in a suitable reaction vial. PCR reactions were performed with 50 pmol primers each added to the reaction vial. Amplifications were carried out using the following profile: one cycle at 95 °C for 10 min, followed by 30 cycles at 95 °C for 30 s, 58 °C for 30 s, and 72 °C for 30 s, followed by 72 °C for 5 min. Lastly, 5 µL of PCR products were run on a 3% of gel dyed with peqGreen.

Colonies of G1-2, G6-7, and G8 showed a yellow colony color, and confirmation of G1-2, G6-7, and G8 identity colonies was carried out by PCR as described below using the single primer pair used to detect the three genotypes. G3-10, G4-9, and G5 genotypes formed white colonies, and their identification was carried out by multiplex using the three primer pairs. DNA from 75 white colonies with suitable dilutions (> 150 colony-forming units (CFU) and < 300 CFU/agar) was extracted as described above. Abundances of white and yellow genotypes were calculated per gram of cecal sample. The same procedure was followed for the original batch of Symbioflor 2^®^ product which was used to inoculate the gnotobiotic mice.

### 2.5. Isolation of RNA from Intestinal Tissue and Preparation of cDNAs

Ileal and colonic RNA extraction was performed using the Mirvana kit (Ambion/ThermoFisher, Waltham, MA, USA) according to the manufacturer’s instructions. RNA quality was determined using the RNA 6000 Nanoassay kit containing chips designed to separate nucleic acid fragments based on their size. The RNA integrity number (RIN) was calculated following electrophoresis of the samples on an Agilent 2100 Bioanalyzer. All tested RNA samples had an RIN above 8.5, indicating that they were all of high quality. RNA concentration A260/A280 and A260/A230 ratios were measured using a Nanodrop 2000 spectrophotometer. Both measures indicate nucleic acid purity. Samples were stored at −80 °C.

Total RNA (2 µg) was converted to single-stranded complementary DNA (cDNA) using the High-Capacity cDNA Reverse Transcription Kit (ThermoFisher, Waltham, MA, USA) according to the manufacturer’s instructions. Real-time PCR of the reference gene, glyceraldehyde-3-phosphate dehydrogenase (*gapdh*), was performed to verify cDNA quality using assay number Mm99999915_g1, designed by Applied Biosystem. Cq values were similar between all groups and in the range of previous values for GAPDH detection, indicating good cDNA synthesis.

### 2.6. Design of Ileal and Colonic Custom TaqMan^®^ Array Card

We designed two custom TaqMan^®^ Array Cards to investigate the expression of 220 genes plus four additional reference genes (*actb*, *gapdh*, *ubc*, *tpb*) for the ileum and the colon as described previously [16]. The genes included in both array cards are listed in Appendix A. We selected genes to investigate the various functions of ileal and colonic cells: those involved in the regulation of cell proliferation and differentiation, cellular signaling, detoxification, DNA-damage detection, growth factors, the inflammasome, inflammatory and immunological responses, the intestinal barrier, lipid synthesis, metabolism, oxidative stress, pattern recognition, and solute transport. The genes were chosen based on published data and the National Center for Biotechnology Information (NCBI) public repository Gene Expression Omnibus.

### 2.7. Gene Expression Experiments

For each customized plate, one different cDNA sample of each of the 10 groups of interest (GF, G1/2, G3/10, G4/9, G5, G6/7, G8, Nissle, Symb, Cv) was loaded and screened against the 224 genes. In addition, the same sample was systematically loaded in each TaqMan OpenArray and used to check the reproducibility between the plates. The cDNA (10 µL) was mixed with TaqMan OpenArray Real-Time PCR Master Mix and loaded onto the cards using the AccuFill™ System. The cards were cycled in an OpenArray NT Cycler System (Applied Biosystems, Waltham, MA, USA) at the integrative microgenomic platform (@Bridge platform, INRAe, Jouy-en-Josas, France).

### 2.8. Gene Expression Analysis and Statistics

Data were extracted using OpenArray Real-Time qPCR Analysis software (Applied Biosystems, Waltham, MA, USA). The fold-change in gene expression (Rq or relative quantification) was calculated using the comparative 2^−ΔΔCq^ method with global normalization of all gene expression data using GenEx software (Multid Analyses, Gothenburg, Sweden). Rq was calculated using the GF group as a control.

The R statistics environment was used for data analyses. Prior to principal component analysis (PCA), we carried out a nonspecific filtering using a one-way permutation test (oneway_test, R package “coin”) to remove genes in the datasets for which the variation in expression was the least informative (threshold set at *p*-value < 0.01). PCA was then carried out on the filtered dataset using the R packages FactoExtra and FactoMineR. The number of genes included in the PCA analysis is specified in the figure legends for each comparison. To define genes expressed differently between the groups, datasets were firstly pre-filtered (one-way analysis of variance (*p* < 0.1) and multivariate comparison of the Rq values was applied by using the Mann–Whitney test (*wilcox.test R function*) using the false discovery rate correction on the *p*-value. A corrected *p*-value <0.05 was considered statistically significant.

### 2.9. Single Real-Time Quantitative PCR Analyses of Gene Expression

Single real-time quantitative PCR assays were used to confirm the results obtained on the TaqMan Open Arrays system using the corresponding TaqMan assay. All gene expression results are expressed with the 2^−∆∆Ct^ method (Rq), using *gapdh* as the housekeeping gene and values from the GF calibrator. Non-parametric Kruskal–Wallis and Dunnett’s range tests were used to determine any significant differences between GF and inoculated groups of mice and *p*-value was corrected for multiple comparisons using statistical hypothesis testing (GraphPad prism software, San Diego, CA, USA). A corrected *p*-value < 0.05 was considered significant.

### 2.10. Availability of Data and Materials

The datasets used and/or analyzed during the current study are available from the corresponding author on reasonable request.

## 3. Results

### 3.1. The Ileal and Colonic Gene Expression Differs according to Which Genotype of Symbioflor2^®^ Is Used

We carried out gene expression profiling by high-throughput open-array qPCR of the mice’s ileum and colon using two customized gene-expression array plates. One was designed for the ileum and the other for the colon, with a selection of candidate genes mainly associated with mucosal defense of the intestine (Appendix A). First of all, we compared the impact of each genotype of Symbioflor2^®^ by carrying out PCA on gene expression profiles in the ileum (Figure 1A) and in the colon (Figure 1B). PCA clustered the different groups of mice according to the genotype they were inoculated with. In the ileum, a clear separation is observed along PC1 between the groups mono-associated with G1/2, G3/10, and G4/9 and the groups mono-associated with G5, G6/7, and G8 (Figure 1A). G1/2 mice diverge the least from the GF group. Mice associated with G3/10 and G4/9 are clearly distinct from the GF along PC1. G5, G6/7, and G8 mono-associated groups are distinct from the GF along PC2. In the colon, we can also observe a separation between the G1/2- and G3/10-associated groups, and the G5-, G6/7-, and G8-associated mice (Figure 1B). The groups that differ the most from the GF are the G5, G6/7, and G8 mono-associated mice, whereas the G4/9 group overlaps that of the GF. Our findings also show that Symbioflor2^®^ genotypes can induce the expression of genes that play a key role in intestinal homeostasis including that of enzymes involved in the turnover of reactive oxygen species, antimicrobial peptides, the intestinal barrier, solute transport, and immune responses (Appendix A). We checked that all the *E. coli* genotypes of Symbioflor2^®^ colonize the intestinal tract at high and similar levels in monoxenic mice. Indeed, 21 days after inoculation, each of the six strains of *E. coli* colonized the caecum of germ-free mice at high levels (10^10^–10^11^ CFU per gram of cecal content) with no difference between strains (Appendix A).

### 3.2. The Intestinal Gene Expression Profile of Mice Associated with the Symbioflor2^®^ Preparation Is Distinct to That of GF but There Is No Predominance of Any Single Genotype

We then explored the effect of the product Symbioflor2^®^ compared to the effect of each single individual genotype in the ileum (Figure 2A) and in the colon (Figure 2B). Data reveal that the Symbioflor2^®^-associated group diverges from GF. However, the impact of the product is not driven by one of the genotypes, whether in the ileum (Figure 2A) or in the colon (Figure 2B). Rather, the impact of the product seems to reflect a balanced host intestinal response. Indeed, data obtained from mice inoculated with Symbioflor2^®^ show that the mix of the six genotypes targets a core set of genes involved in defense mechanisms, including oxidative stress, anti-microbial peptide production, solute transport, and immune responses (Appendix A). Concomitantly, colonization data reveal that no genotype predominates in the gut of Symbioflor2^®^-associated mice (Figure 3). Indeed, G1/2, G6/7, and G8 groups (that cannot be distinguished given their high genomic similarity), as well as G3/10, colonize the intestine well (Figure 3). G4/9 and G5 abundances are more variable between mice but reach good colonization levels in four mice (~10^8^ CFU/g of cecal content).

### 3.3. The Ileal and Colonic Gene Expression Profiles of Mono-Associated Mice Converge toward the CV Profile

Gene expression profiles in mono-associated mice were then compared to those obtained for GF and CV mice. As expected, this analysis firstly revealed a marked difference between CV and all the other groups, in the ileum (Figure 4A) and in the colon (Figure 4B). However, we observed a tendency for all groups of mono-associated mice to converge toward CV mice. In particular, in the ileum, mice mono-associated with G5, G6/7, and G8 separate from GF mice and shift toward the CV group. In the colon, we find the same tendency in the G1/2, G3/10, and G8 mono-associated mice.

### 3.4. The Nissle Strain and the Six Bacterial Genotypes of Symbioflor2^®^ Have Different Effects on Intestinal Gene Expression Profile, Particularly in the Colon

We then compared the ileal and colonic gene expression profile of mice mono-associated with each of the six genotypes of Symbioflor2^®^ and the Nissle strain. In the ileum, we observed that the gene expression profile of mice mono-associated with Nissle is closer to the G4/9 and G1/2 than all the other strains (Figure 5A). In contrast, it clearly diverges from that of the G5, G6/7, and G8 mono-associated mice (Figure 5A). In the colon, the gene expression profile of mice mono-associated with Nissle is distinct from that found in mice associated with the genotypes of Symbioflor2^®^ (Figure 5B). The gene expression profiles that are the most dissimilar from Nissle associated mice are those associated with G1/2 and G3/10 (Figure 5B). These divergences between Nissle and the six genotypes of Symbioflor2 are not linked to a difference in colonization levels (Appendix A).

### 3.5. The Six Bacterial Strains of Symbioflor^®^ 2 Modulate Genes Involved in Several Cell Functions

We performed a multivariate statistical test to identify genes whose expression was significantly different between GF and both the mono-associated groups and the Symbioflor2^®^ group. The lists are given in the Appendix A for the ileum and the colon, respectively. In the ileum, we found that 13, 35, 21, 29, 36, and 32 genes were differentially expressed between the GF and the G1/2, G3/10, G4/9, G5, G6/7, and G8 mono-associated mice. Nine genes differed between the GF and the Symbioflor2^®^-associated mice. In the colon, 16, 24, six, 27, 28, and 29 genes were differentially expressed between the GF and the G1/2, G3/10, G4/9, G5, G6/7, and G8 mono-associated mice. Eleven genes differed between the GF and the Symbioflor2^®^-associated mice. The genes involved in ROS/RNS turnover, anti-microbial peptide production, immune responses, detoxification, and mucosal fortification are altered by inoculation of all genotypes. The gene expression of several solute transporters, involved in ionic transport, and possibly luminal environment regulation, are also modulated by the strains.

Single real-time quantitative PCR assays were used to confirm the results obtained on the TaqMan Open Arrays system using the corresponding TaqMan assay (Figure 6 in the ileum and Figure 7 in the colon). The data confirmed that the Symbioflor2^®^ genotypes increase the expression of several genes involved in mucosal defense in the ileum (Figure 6) and in the colon (Figure 7). In the ileum, it is shown for genes associated with (i) ROS/RNS turnover: *duox2* (dual oxidase 2), *duoxa2* (dual oxidase activator 2), and *nos2* (nitric oxide synthase 2), (ii) antimicrobial peptide production: *reg3β, reg3γ* (regenerating islet-derived β and γ), and *ang4* (angiogenin-4), (iii) mucosal fortification: *alpi* (alkaline phosphatase, intestinal) and *cldn4* (claudin 4), (iv) immune response: *rorc* (RAR-related orphan receptor gamma), and (v) ionic water transport: *aqp3* (aquaporin 3). In the colon, it is shown for genes involved in (i) ROS turnover: *duox2* and *duoxa2*, (ii) *pl2ga2* (secretory phospholipase A group IIA), (iii) cell proliferation: *ccne1* (cyclin E1), (iv) ionic transport: *clca4* (Ca^2+^-activated chloride channel 4), (iv) immune response: *rorc*, *thpo* (thrombopoietin), *Il6* (interleukin-6), *TNFα* (tumor necrosis factor α), and *IL17d*.

## 4. Discussion

Symbioflor2^®^ was previously shown to have beneficial effects on IBS in humans. The objective of this study was to deepen our knowledge of this probiotic product which contains six *E. coli* genotypes. We explored the effect of the six genotypes on the gene expression in the ileum and in the colon, both individually and combined in the probiotic product. This analysis was carried out using the simplified model of gnotobiotic mice. We showed dynamic transcriptional responses, both in the ileum and in the colon, to the six *E. coli* genotypes included in Symbioflor2^®^. This transcriptomic study revealed distinct profiles for the different mono-associations. Indeed, in the ileum and in the colon, we found a clear separation in the gene expression profile between the G1/2, G3/10, and G4/9 groups and the G5, G6/7, and G8 mono-associated mice. The most pronounced effects were obtained with the genotypes G3/10 and the genotypes G5, G6/7, and G8, whereas the G4/9 and G1/2 transcriptomic-associated profiles diverge the least from the GF values. Interestingly, two of the genotypes which have the strongest effect, G6/7 and G8, are able to colonize the human gut for several weeks [17].

We show that Symbioflor2^®^ genotypes increase the expression of the genes that play a key role in mucosal defense and in the maintenance of the symbiotic host–microbiota relationship. A common intestinal response to all Symbioflor2^®^ genotypes is the increased expression of *duox2* and its activator *duoxa2*. This capacity to increase these two gene expressions is also shared with a commensal strain of *E. coli* (CEC15) [16], previously isolated from suckling rodents [18], and with the probiotic Nissle strain [16]. Several studies showed that Duox-generated ROS play a pivotal role in the innate intestinal defense response and in regulating the homeostasis of the gut bacterial community. This process was preserved throughout evolution and was demonstrated in insects [19,20] and nematodes [21]. *Duox* gene expression is upregulated in humans in several intestinal diseases, such as irritable bowel syndrome [22] and intestinal inflammation [23,24], which could suggest it plays a detrimental role. However, this is debated as, in a disease setting, the upregulation of *DUOX2* could be a compensatory mechanism that could substitute for a defective host defense response [25]. Furthermore, it was shown that the nox/duox pathway is activated by bacterial ligand, particularly in goblet cells residing on the top of the intestinal epithelium. The ROS that are produced trigger luminal MUC2 secretion that favors the clearing of bacteria from the crypt opening, thereby protecting the lower crypt and intestinal stem cells from bacterial intrusion [26]. These previous findings identify DUOX2 as a critical modulator in mutualistic host–microbiota interactions, fundamental in maintaining gut immune homeostasis. In our study, the increase of *duox* and *duoxa2* gene expression by Symbioflor2^®^ genotypes is below CV levels. This suggests that levels obtained in mono-associated mice are within the range of physiological values. Interestingly, in the ileum, we showed that G1/2, G3/10, and G4/9 also increase the expression of *aqp3*. It was recently shown that AQP3 transports H_2_O_2_ generated at the cell surface by NOX1 and DUOX2 to mediate signal transduction in colonic epithelia [27], reinforcing the assumption that *E. coli* strains modulate ROS turnover and signaling.

In addition to Duox2 and Duoxa2, Symbioflor2^®^ genotypes can induce the expression of a panel of genes involved in intestinal defense mechanisms. This is the case for genes encoding anti-microbial peptides. In fact, we show, in the ileum, that Reg3γ is targeted by G1/2 and G3/10, Reg3β by G1/2, G3/10, G5, and G8, and ang4 by G3/10, G4/9, G5, and G6/7. In the colon, Reg3γ is targeted by G5, G6/7, and G8 and the Symbioflor2^®^ product itself. A previous study reveals that fecal hBD-2 peptide increases in healthy subjects after a three-week treatment with Symbioflor2^®^. This impact of genotypes was also confirmed by in vitro studies revealing that at least one of the Symbioflor2^®^ genotypes is able to increase intestinal hBD-2 gene expression in a way that is comparable to probiotic *E. coli* Nissle 1917 [13]. Our work builds on this previous study showing that Symbioflor2^®^ genotypes modulate a large spectrum of anti-microbial peptides.

Symbioflor2^®^ genotypes are also able to target genes involved in ileal mucosal reinforcement as shown with the increase of *cldn4* and *alpi* gene expression in mono-associated mice. Several studies showed that alpi plays a beneficial role in maintaining healthy symbiotic microbiota–host relationships. In zebrafish, it contributes to the detoxification of lipopolysaccharide (LPS), thereby minimizing the pro-inflammatory potential of this bacterial product [28], and, in mice, alpi deficiency results in more severe colitis in a chemically induced disease model [29]. Interestingly, our data also suggest that, at the immune response level, the colon is highly responsive to Symbioflor2^®^ genotypes. Symbioflor2^®^ genotypes are seen to increase the expression of genes involved in immune response, such as *IL6*, *IL17d*, or *RORc*, whereas their expression is not modified in GF and CV mice. We also observed that the genes encoding immunoregulatory cytokines (*Il6*, *TNF*, *Il17d*) are activated by the Symbioflor2^®^ genotypes but mainly in the colon. Here again, this either implies that the colon is immunologically more responsive to these strains than the ileum or reflects the fact that *E. coli* colonization is higher in the lower part of the digestive tract [30].,

We showed that the six bacterial genotypes of Symbioflor2^®^ have different impacts on intestinal gene expression profile. All Symbioflor2^®^ genotypes originated from the fecal sample of a healthy volunteer, and genomic and proteomic comparison of the six genotypes revealed that there is a relatively low genetic diversity between them. G6/7 and G8 are the most closely related, as few differences were detected between the genome sequences and the proteomes. G1/2 is also more closely related to the G6/7 and G8 genotypes than to the others, whereas G3/10 was the least similar to the other genotypes [31,32]. The transcriptomic profiles are consistent with the genomic differences between G6/7, G8, and G3/10 because the transcriptomic profiles of G6/7- and G8-associated mice overlap both in the ileum and in the colon and diverge from the G3/10 mono-associated mice. However, although the genomic content of G1/2 is more closely related to G6/7 and G8 than the other strains, the transcriptomic profile is distinct from these two genotypes and more similar to G4/9. The G5-associated profile generally overlaps with those associated with G6/7 and G8. G4/9 has one of the lowest impacts on the intestinal gene expression profile, which could be linked to its small genome size. Thus, from our study, we can partially match the transcriptomic intestinal profiles of each genotype to their genomic similarity. Interestingly, the six bacterial genotypes and the Nissle strain have a different impact on the intestinal gene expression profile. This is particularly clear at the colonic level. This suggests that these probiotics are not redundant and can each exert an independent beneficial effect on gut health.

## 5. Conclusions

In conclusion, we found that, individually, each of the six bacterial genotypes of Symbioflor2^®^ modifies genes involved in several cell functions, including those associated with mucosal defense. Thus, for example, the expressions of *duox2* and *duoxa2*, which are involved in innate intestinal defense response, and of *cldn4* and *alpi*, involved in mucosal reinforcement and of immunoregulatory cytokines, are increased in mice mono-associated with bacterial genotypes of Symbioflor2^®^. When the product was given to mice, we observed that no single genotype was predominant. This revealed that the balance between strains established in the preparation is preserved upon ingestion and leads to a balanced combination of responses from the host. Altogether, our work contributes to a deeper understanding of the beneficial effects of probiotic *E. coli* strains, which could benefit targeted therapeutic strategies.

## Figures and Tables

**Figure 1 microorganisms-08-00512-f001:**
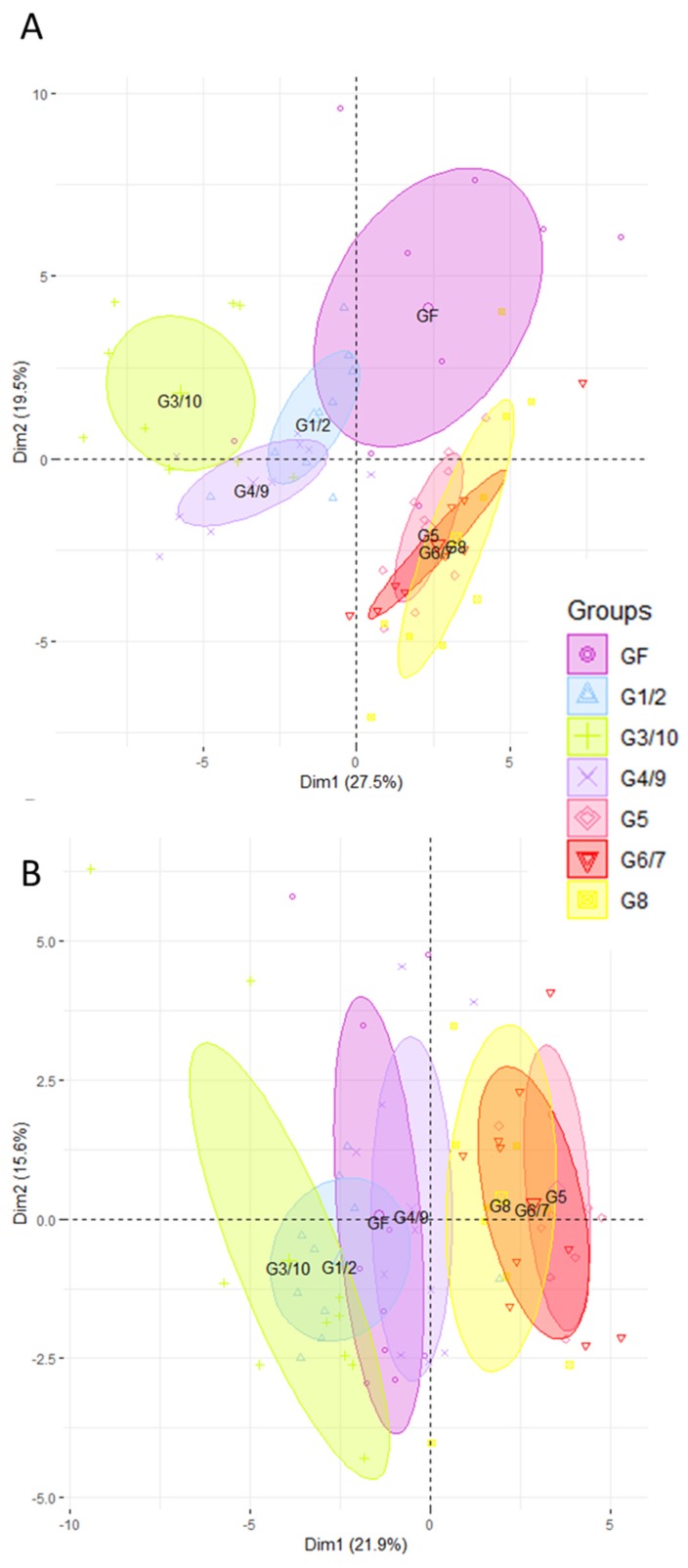
The ileal and colonic gene expression differs according to which genotype of Symbioflor2^®^ is administered. Ileal (**A**) and colonic (**B**) gene expression was analyzed in mice mono-associated with one of the six Symbioflor2^®^ genotypes (G1/2, G3/10, G4/9, G5, G6/7, G8) and in germ-free (GF) mice. Rq values that represent the abundance of transcripts relative to GF group were used for principal component analysis (PCA) after trimming datasets for the least informative genes (see Section 2). In total, 56 genes were retained for PCA in the ileum and 40 in the colon; *n* = 10 for each group of mice.

**Figure 2 microorganisms-08-00512-f002:**
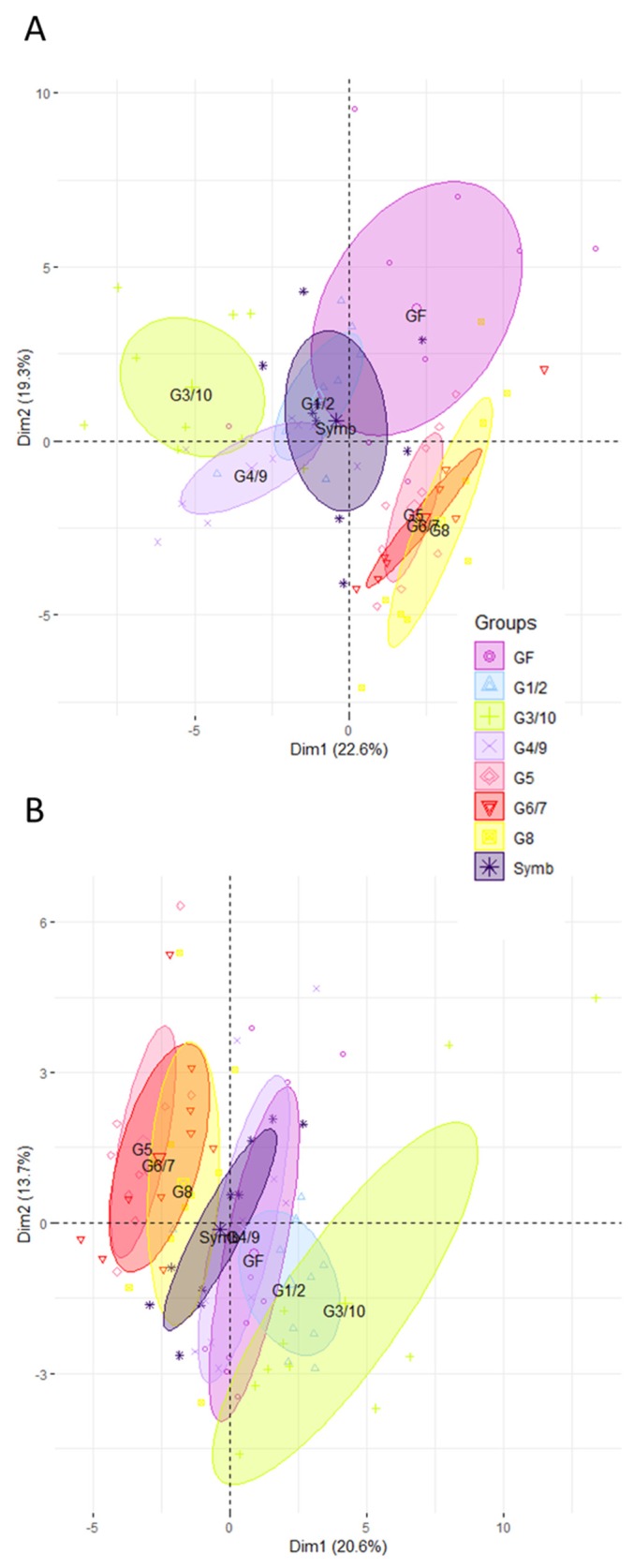
The impact of the Symbioflor2^®^ product on intestinal gene expression is not driven by any single genotype. Ileal (**A**) and colonic (**B**) gene expression was analyzed in mice mono-associated with one of the six Symbioflor2^®^ genotypes (G1/2, G3/10, G4/9, G5, G6/7, G8), in mice inoculated with the Symbioflor2^®^ product containing the six genotypes (Symb), and in GF mice. Rq values that represent the abundance of transcripts relative to GF group were used for PCA analysis after trimming datasets for the least informative genes (see Section 2). In total, 50 genes were retained for PCA in the ileum and 45 in the colon; *n* = 10 for each group of mice.

**Figure 3 microorganisms-08-00512-f003:**
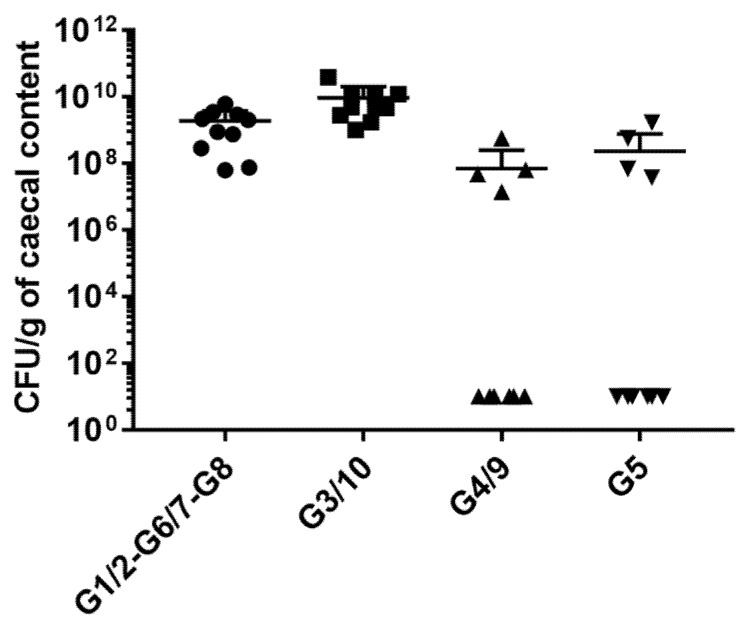
All Symbioflor2^®^ genotypes colonize the intestine in Symbioflor2^®^-associated mice. The colonization level of each of the *E. coli* genotypes was quantified by cultivation and subsequent PCR on grown colonies obtained from the cecal content of gnotobiotic mice inoculated with Symbioflor2^®^. Cecal samples were plated on nutrient medium and cultivated aerobically. Colonies of G1-2, G6-7, and G8 showed a yellow colony color, and their identities were confirmed by PCR using the single primer pair used to detect the three genotypes (G1-2, G6-7, and G8 were not distinguishable by PCR due to the genomic similarities between the three genotypes). G3-10, G4-9, and G5 genotypes formed white colonies, and their identification as G3-10, G4-9, and G5 colonies was carried out by multiplex using the three specific primer pairs. Absolute abundances of white and yellow genotypes were calculated per gram of cecal sample.

**Figure 4 microorganisms-08-00512-f004:**
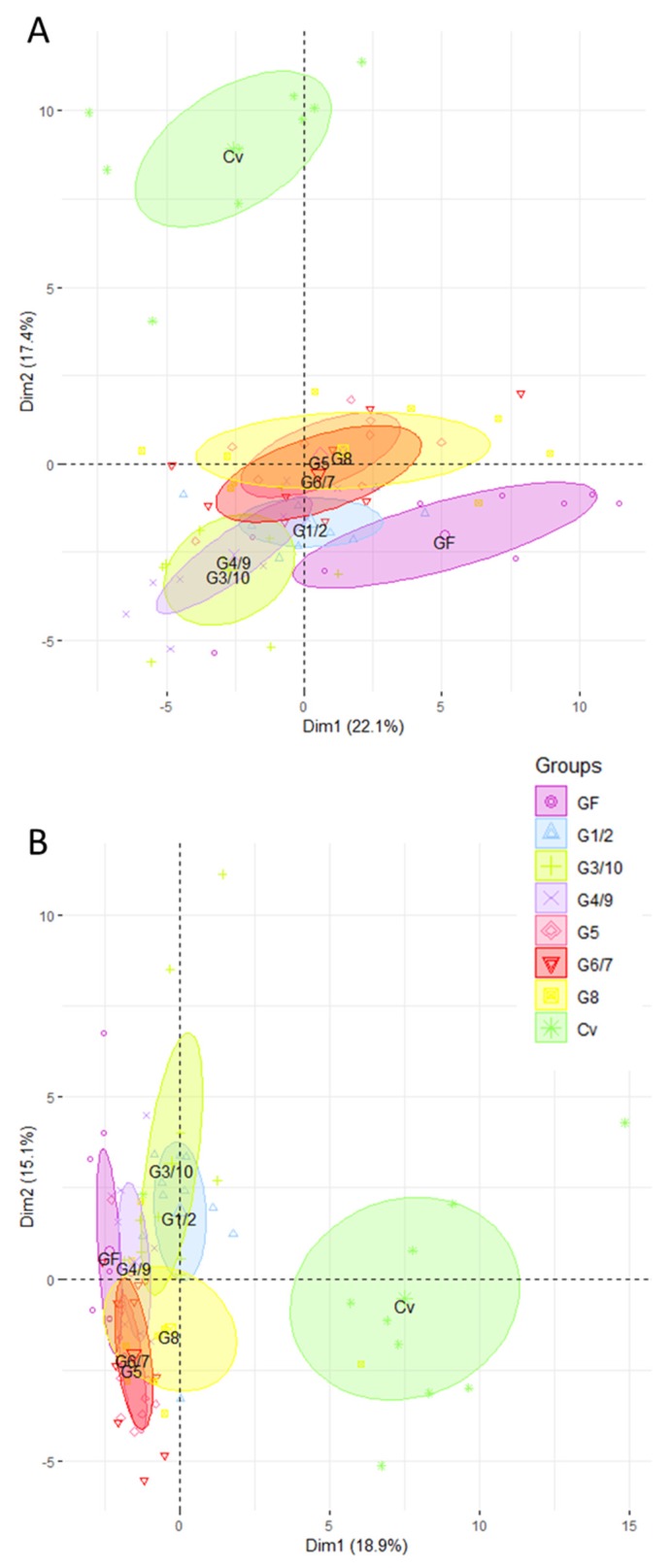
The intestinal gene expression profiles of mono-associated mice converge toward the CV profile. Ileal (**A**) and colonic (**B**) gene expression was analyzed in mice mono-associated with one of the six Symbioflor2^®^ genotypes (G1/2, G3/10, G4/9, G5, G6/7, G8), in CV and in GF mice. Rq values that represent the abundance of transcripts relative to GF group were used for PCA analysis after trimming datasets for the least informative genes. In total, 84 genes were retained for PCA in the ileum and 60 in the colon; *n* = 10 for each group of mice.

**Figure 5 microorganisms-08-00512-f005:**
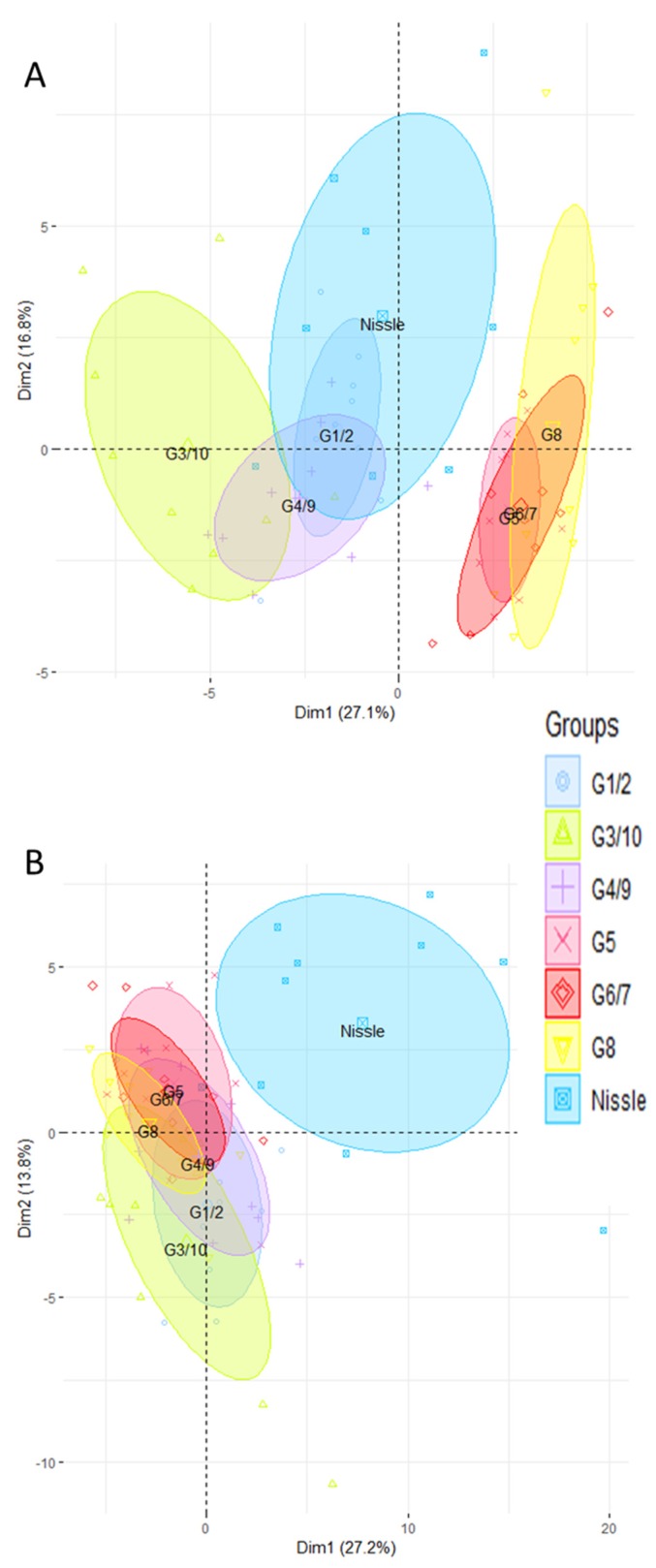
The intestinal gene expression profile of Nissle mono-associated mice is distinct from that of Symbioflor2^®^ genotypes mono-associated mice, particularly in the colon. Ileal (**A**) and colonic (**B**) gene expression was analyzed in mice mono-associated with one of the six Symbioflor2^®^ genotypes (G1/2, G3/10, G4/9, G5, G6/7, G8), or with the Nissle *E. coli* strain and in GF mice. Rq values that represent the abundance of transcripts relative to GF group were used for PCA analysis after trimming datasets for the least discriminant genes. In total, 49 genes were retained for PCA in the ileum and 79 in the colon; *n* = 10 for each group of mice.

**Figure 6 microorganisms-08-00512-f006:**
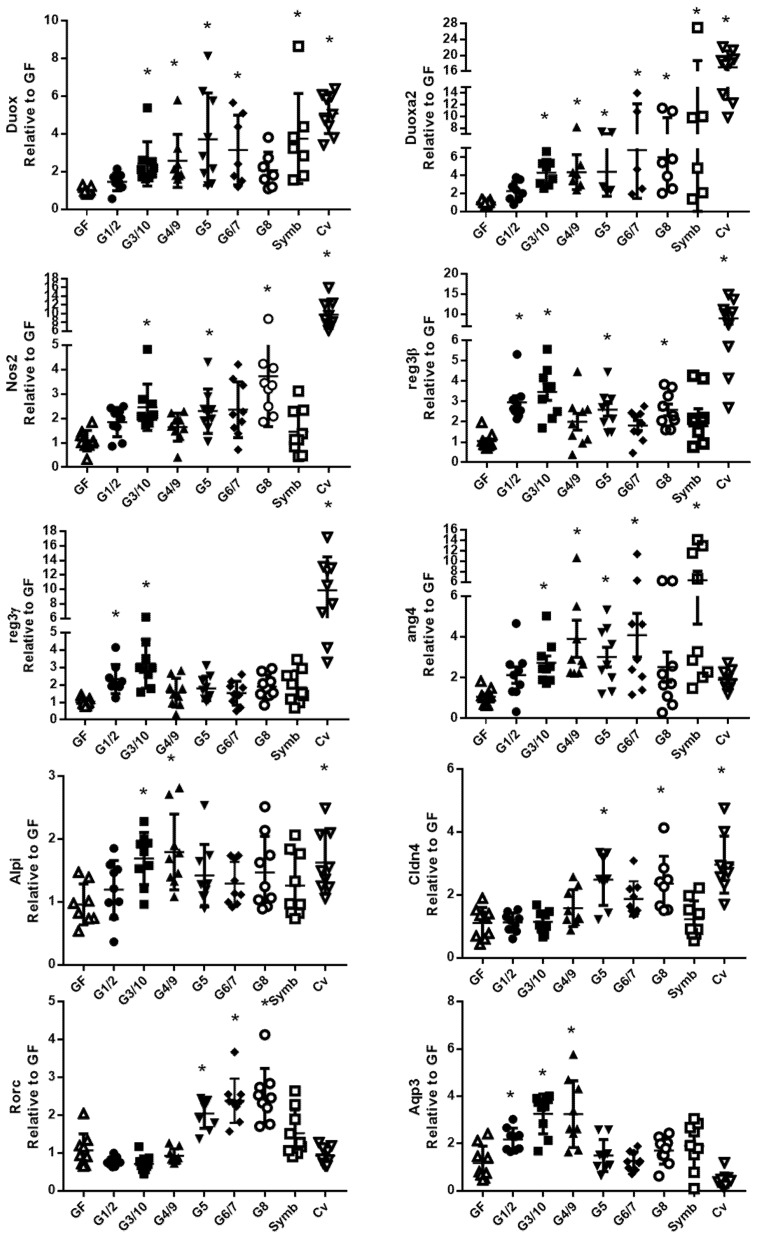
The six bacterial strains of Symbioflor2^®^ modulate genes involved in several cell functions in the ileum. Relative gene expression was measured with single Taqman Assays for *duox2*, *duoxa2*, *nos2*, *reg3β*, *reg3γ*, *ang4*, *alpi*, *cldn4, rorc*, and *aqp3* in the ileum of mice mono-associated with one of the six Symbioflor2^®^ genotypes (G1/2, G3/10, G4/9, G5, G6/7, G8), inoculated with the Symbioflor2^®^ product that contains the six genotypes (symb), in GF mice and in Cv mice; *n* = 9–10 mice/group. All values are presented as the means ± standard error of the mean (SEM). * indicates a statistical difference compared to GF mice (adjusted *p*-value <0.05).

**Figure 7 microorganisms-08-00512-f007:**
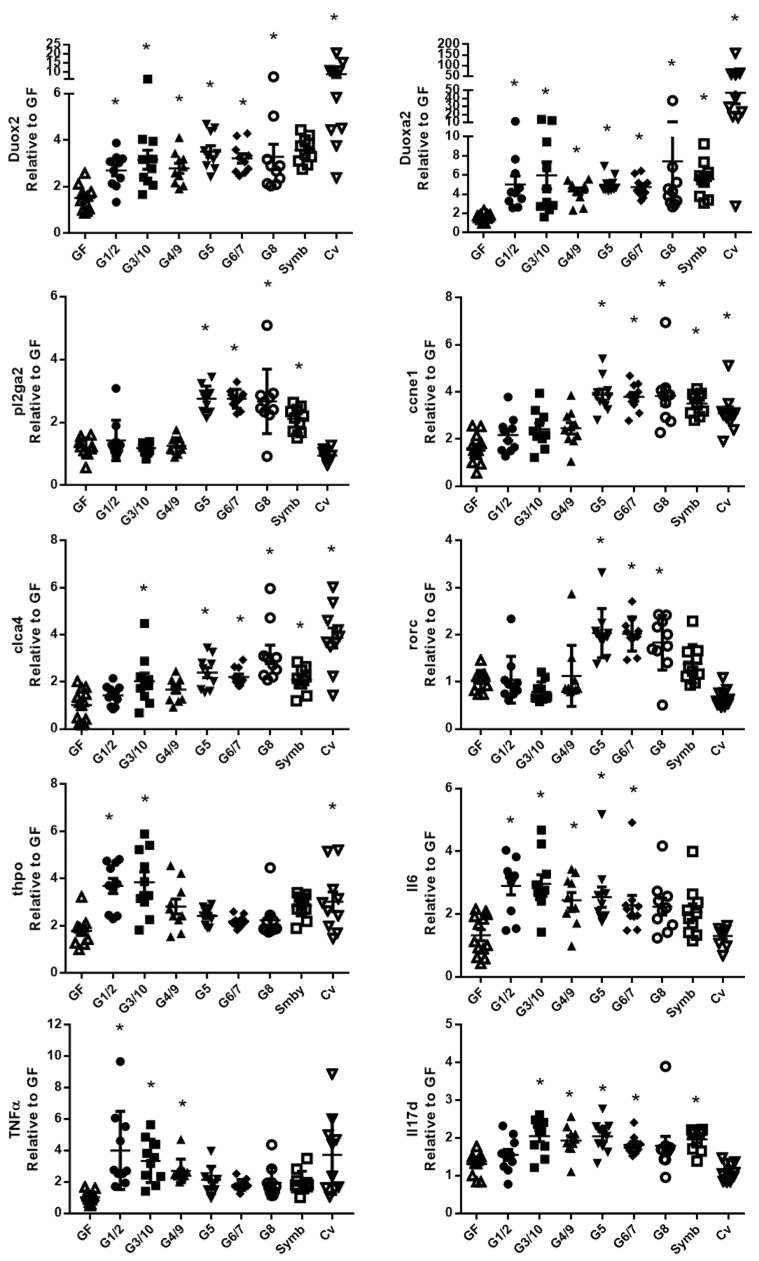
The six bacterial strains of Symbioflor2^®^ modulate genes involved in several cell functions in the colon. Relative gene expression was measured with single Taqman Assays for *duox2*, *duoxa2*, *pl2ga2*, *ccne1*, *clca4*, *rorc*, *thpo*, *Il6*, *TNFα,* and *IL17d* in the colon of mice mono-associated with one of the six Symbioflor2^®^ genotypes (G1/2, G3/10, G4/9, G5, G6/7, G8), inoculated with Symbioflor2^®^ product that contains the six genotypes (symb), in GF mice and in Cv mice; *n* = 9–10 mice/group. All values are presented as the means ± SEM. * indicates a statistical difference compared to GF mice (adjusted *p*-value <0.05).

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
