# Peer review of "Symbioflor2® Escherichia coli Genotypes Enhance Ileal and Colonic Gene Expression Associated with Mucosal Defense in Gnotobiotic Mice"

_microorganisms, 2020, doi:10.3390/microorganisms8040512_

Round 1

Reviewer 1 Report

The authors analysed the impact of each of the six E. coli components of the probiotic Symbioflor2 on host intestinal transcriptional responses using the gnotobiotic mice model. The effects were compared with those mediated by the whole Symbioflor2 formula and the probiotic E. coli Nissle 1917 as well. Microarray analysis included 220 selected genes involved in several processes such as cell proliferation, cellular signalling, defence, intestinal barrier, metabolism, oxidative stress and immune responses. Results showed that ileum and colonic transcriptional profiles were different for each probiotic strain and that Symbioflor elicited a balanced response that combines the single genotype-mediated effects.

Nowadays, clinical applications of probiotics are receiving substantial attention. However, there is still much to learn about the intrinsic mechanisms of probiotic action. Thus, new research is required to provide basic knowledge on probiotic-host interactions, a key issue that will help to develop optimal probiotic-based strategies for therapeutic purposes. In this context, the manuscript addresses a relevant study aimed at deciphering the molecular mechanisms involved in the beneficial effects of the commercial probiotic formula Symbioflor, whose effectiveness in reducing IBD symptoms has been evidenced in clinical trials.

The study is well designed, conducted and justified. The methodology is adequate, and results support the main conclusions.

However, some issues should be addressed to clarify specific points and to improve the outcome.

1.       Lines 153-155: Authors state that selection of genes was done based on published data and the NCBI repository Gene Expression Omnibus. More information on the rational for the selection should be given with more detail. Are these genes relevant for normal gut function? Are there altered in IBD?  Why these genes and not others? For instance, I have not seen in supplemental tables 2-3 genes encoding the tight junction proteins claudin-1 or ZO-1, both commonly analysed as intestinal barrier markers. Also, I couldn’t find the gene encoding the ortholog human beta-defensin 2, an antimicrobial peptide shown to be upregulated by Symbioflor in in vitro studies and healthy individuals (discussed in lines 303-307).

2.        Germ-free mice represent a simplified model system to study the effect of gut microbes on host gastrointestinal function and physiology. In addition, it allows to compare the effects between mono-colonized mice and mice colonized with a combination of bacteria. However, this model does not reflect the impact of normal microbiota on the probiotic effects. Have authors tried to analyse the effects of Symbioflor in conventional mice or in experimental models of colitis? This last model would be especially relevant to evaluate the ability of Symbioflor and its single E. coli genotypes in ameliorating IBD symptoms and in counteracting altered gene expression of intestinal barrier and inflammatory mediators.

3.       Figures 6-7. Could authors explain why the expression level of the immune markers Rorc (ileum and colon) and IL6 and IL17d (colon) in conventional mice approximately coincide with that of germ-free mice? In contrast, expression of these genes is increased by either Symbioflor or some of its bacterial components. Comments about this issue should be included in the discussion section.

4.       Methods: Were conventional mice seven to eight-week-old males as germ-free mice?

5.       Line 89: Please indicate the growth medium used for bacterial cultures.

6.       Line 92:  Did mice receive a single-dose inoculation?

7.       Section 3.4: Were the colonization levels of Nissle 1917 close to that of Symbioflor single E. coli strains? Please, give Nissle colonization data either in the text (CFU/g of caecal content) or included in supplemental figure 1.   

8.       Supplemental Tables 4-5: Why are some genes indicated in red? What does the difference mean? Please explain.

9.       Line 314 (spelling mistake): The name of il6 should start with a capital letter (I).

Author Response

  1. Lines 153-155: Authors state that selection of genes was done based on published data and the NCBI repository Gene Expression Omnibus. More information on the rational for the selection should be given with more detail. Are these genes relevant for normal gut function? Are there altered in IBD?  Why these genes and not others? For instance, I have not seen in supplemental tables 2-3 genes encoding the tight junction proteins claudin-1 or ZO-1, both commonly analysed as intestinal barrier markers. Also, I couldn’t find the gene encoding the ortholog human beta-defensin 2, an antimicrobial peptide shown to be upregulated by Symbioflor in in vitro studies and healthy individuals (discussed in lines 303-307).

The list of genes given in supplemental tables 2-3 has now been revised to clarify the function of each of them. As specified in the Materials and Methods section, we selected genes involved in the various functions of ileal and colonic cells (epithelial cell renewal, detoxification, DNA-damage detection, growth factors, inflammasome, inflammatory and immunological responses, intestinal barrier, lipid synthesis, metabolism, oxidative stress, pattern recognition, and transport). By these choices, our objective was to have a global view of the impact of Symbioflor2 and its genotypes on host intestinal physiology and to identify potential new targets for probiotics. Among them, several sets of genes were relevant to intestinal inflammation such as, for example, those involved in inflammasome, inflammatory and immunological responses, intestinal barrier and pattern recognition.

We totally agree with the referee that investigating intestinal barrier markers is relevant to our study. With this aim we included several physical and chemical intestinal barrier markers at the ileum and colonic level, taking into account the specificity of both intestinal sites. For example, we included claudins (7 for the ileum, 5 for the colon), occludin, cadherin and several types of mucins (secreted or transmembrane mucins: 4 for the ileum and 6 for the colon) (please see supplemental tables 2-3). At the ileum level, we investigated ZO1 and ZO2 (Tjp1 and Tjp1) but we found no difference between groups. 

4 types of β-defensin were included in colonic taqMan array card. We found that none of the strains was able to modulate these β-defensins, except for the G5 genotype, which is able to modulate Def2 and Def8 expression. We cannot exclude the possibility that the genotypes have an impact on β-defensin at the ileum level. However, as specified below, we decided to build on prior knowledge of the impact of the strains by using a larger repertoire of anti-microbial peptides (14 for the ileum and 18 for the colon). Please note that we have changed Lines 325 and 329-330 accordingly.

  1. Germ-free mice represent a simplified model system to study the effect of gut microbes on host gastrointestinal function and physiology. In addition, it allows to compare the effects between mono-colonized mice and mice colonized with a combination of bacteria. However, this model does not reflect the impact of normal microbiota on the probiotic effects. Have authors tried to analyse the effects of Symbioflor in conventional mice or in experimental models of colitis? This last model would be especially relevant to evaluate the ability of Symbioflor and its single E. coli genotypes in ameliorating IBD symptoms and in counteracting altered gene expression of intestinal barrier and inflammatory mediators.

As highlighted by the referee, we carried out this study on germ-free mice because this represents a simplified model system allowing us to see the specific impact of bacterial strains. It was particularly useful in our study to decipher the impact of closely-related strains. We do agree with the referee that this line of research would be very interesting to pursue, according to the data we obtained from gnotobiotic mice. We have not carried out these experiments yet but there is always a possibility of completing our work using colitis models.

  1. Figures 6-7. Could authors explain why the expression level of the immune markers Rorc (ileum and colon) and IL6 and IL17d (colon) in conventional mice approximately coincide with that of germ-free mice? In contrast, expression of these genes is increased by either Symbioflor or some of its bacterial components. Comments about this issue should be included in the discussion section.

We now comment on the points raised by the referee (Please see lines 447-454).

  1. Methods: Were conventional mice seven to eight-week-old males as germ-free mice?

Gnotobiotic mice were euthanized at eleven to twelve weeks (arrival at our gnotobiotic facility around seven to eight + one week without any disturbance + 3 weeks post-inoculation). Cv mice were matched with gnotobiotc mice. Details regarding Cv are now given in the Materials and Methods section: please see lines 68-73 and lines 103-105.

  1. Line 89: Please indicate the growth medium used for bacterial cultures.

Strains were grown in LB (Luria Bertani) medium (please see line 94).

  1. Line 92:  Did mice receive a single-dose inoculation?

Yes, in all cases, initially GF mice were inoculated once by oral gavage. This is now specified in the Materials and Methods section (please see lines  92-93).

  1. Section 3.4: Were the colonization levels of Nissle 1917 close to that of Symbioflor single E. coli strains? Please, give Nissle colonization data either in the text (CFU/g of caecal content) or included in supplemental figure 1.   

Colonization levels of Nissle 1917 are now added in supplemental Figure 1 (please see also line 253-254).

  1. Supplemental Tables 4-5: Why are some genes indicated in red? What does the difference mean? Please explain.

Missing information is now given in the legend of Supplemental Tables 4-5.

  1. Line 314 (spelling mistake): The name of il6 should start with a capital letter (I).

This has now been corrected (now line 337).

Reviewer 2 Report

In this manuscript, Unai et al. investigated the ileal and colonic transcriptional responses in gnotobiotic mice colonized with Symbioflor2® and each genotype. The gene expression profiles were also compared with conventional mice and mice colonized with probiotic E. coli Nissle 1917. This study is well-designed. I have a few comments which I would like the authors to address.

  1. What’s the rationale of choosing the dose of 108 bacteria/100uL for each genotype and 5 x 107 viable bacteria/mouse? What are the equivalent doses in human?
  2. What’s the rationale of choosing 21-day as the inoculation period?
  3. The authors labeled p-values < 0.05 in Figure 6 and 7. I would appreciate it if the authors provide adjusted p-values instead of raw p-values.
  4. Pathway analysis would be helpful to reveal the different pathways between Symbioflor2® and each genotype.

Author Response

In this manuscript, Unai et al. investigated the ileal and colonic transcriptional responses in gnotobiotic mice colonized with Symbioflor2® and each genotype. The gene expression profiles were also compared with conventional mice and mice colonized with probiotic E. coli Nissle 1917. This study is well-designed. I have a few comments which I would like the authors to address.

  1. What’s the rationale of choosing the dose of 108 bacteria/100uL for each genotype and 5 x 107 viable bacteria/mouse? What are the equivalent doses in human?

1 mL of Symbioflor® 2 formula is composed of bacteria culture containing the Escherichia coli (cells and autolysate) equivalent to 1.5 to 4.5x107 living cells. The recommended daily dose at the onset of treatment contains from 3x107 to 13.5x107 bacteria. We take an average value of 5x107 viable bacteria to inoculate mice to respect the equilibrium beween the strains contained in the initial formula.

The dose of 108 bacteria is routinely used in our lab to obtain monoxenic mice. This dose allows a good colonization of the strain and it is well tolerated by germ free mice.

  1. What’s the rationale of choosing 21-day as the inoculation period?

Several reasons led us to opt for the 21-day inoculation period. Our objective was to investigate the impact of the genotypes and the product in a homeostatic state.

Indeed, several studies have shown that there is a dynamic reprogramming of intestinal responses following the transfer of a complex or simplified microbiota into germfree (GF) animals and that the homeostatic state is reached in the 3 weeks post-inoculation [1, 2]. In accordance, regarding the impact of commensal E. coli strain, our team also showed that a commensal E coli strain elicits sequential epithelial remodeling that can result in a microbiota-compliant state over the first 3 weeks post-inoculation [3].

  1. El Aidy, S.; van Baarlen, P.; Derrien, M.; Lindenbergh-Kortleve, D.J.; Hooiveld, G.; Levenez, F.; Dore, J.; Dekker, J.; Samsom, J.N.; Nieuwenhuis, E.E. et al. Temporal and spatial interplay of microbiota and intestinal mucosa drive establishment of immune homeostasis in conventionalized mice. Mucosal Immunol 2012, 5(5), 567-579.
  2. Tomas, J.; Wrzosek, L.; Bouznad, N.; Bouet, S.; Mayeur, C.; Noordine, M.L.; Honvo-Houeto, E.; Langella, P.; Thomas, M.; Cherbuy, C. Primocolonization is associated with colonic epithelial maturation during conventionalization. FASEB J 2013, 27(2), 645-655.
  3. Tomas, J.; Reygner, J.; Mayeur, C.; Ducroc, R.; Bouet, S.; Bridonneau, C.; Cavin, J.B.; Thomas, M.; Langella, P.; Cherbuy, C. Early colonizing Escherichia coli elicits remodeling of rat colonic epithelium shifting toward a new homeostatic state. ISME J 2015, 9(1), 46-58.

  1. The authors labeled p-values < 0.05 in Figure 6 and 7. I would appreciate it if the authors provide adjusted p-values instead of raw p-values.

Fig. 6 and Fig. 7 show single QPCR data using the same sets of primers as the ones used in open array card. In this case, to determine significant differences between GF and inoculated groups of mice, non parametric Kruskal-Wallis and Dunnett’s range tests were used in conjunction. P-values shown in Fig. 6 and Fig. 7 are adjusted p-values. This is now corrected in the figure legend (please see legend of Fig. 6 and Fig. 7). We now also describe single QPCR  and the statistical tests used to analyze data in the Materials and Methods section (please see lines 186-193). Please also note that we have included adjusted p-values in Sup Table 4 and 5.

  1. Pathway analysis would be helpful to reveal the different pathways between Symbioflor2® and each genotype.

We agree with the referee that it is relevant to associate genes with a cellular pathway. Following a comment of referee 1, we have now described the function of each gene loaded on the open array cards (Please see Sup Table 2 and 3).

In this study, we investigated 220 candidate genes. This represents less than 1-2 % of the mouse genome. In these conditions, we cannot use pipeline dedicated to transcriptomic analysis, such as Ingenuity Pathway Analysis to get function enrichment. Indeed, for these analyses, our dataset is biased (due to the small number of genes and subjective selection of genes in a targeted function). This is why we have specified the cellular pathway associated to each of the genes modulated in our groups (Please see Sup Table 4 and 5).